# Advancing Health and Sustainability: A Holistic Approach to Food Production and Dietary Habits

**DOI:** 10.3390/foods13233829

**Published:** 2024-11-27

**Authors:** Graça P. Carvalho, Eduardo Costa-Camilo, Isabel Duarte

**Affiliations:** 1Bioscience School of Elvas, Portalegre Polytechnic University, 7350-092 Elvas, Portugal; eduardocostacamilo@ipportalegre.pt; 2Research Centre for Endogenous Resource Valorization (VALORIZA), 7300-555 Portalegre, Portugal; 3Applied Biomolecular Sciences Unit (UCIBIO), NOVA School of Science and Technology, Universidade NOVA de Lisboa, 2829-516 Caparica, Portugal; 4Instituto Nacional de Investigação Agrária e Veterinária (INIAV), Apartado 6, 7350-951 Elvas, Portugal; isabel.duarte@iniav.pt; 5GeoBioTec, OVA School of Science and Technology (NOVA FCT), Universidade NOVA de Lisboa, 2829-516 Caparica, Portugal

**Keywords:** healthy food production, sustainable agriculture, sustainable food systems, food literacy, local food

## Abstract

Producing healthier food requires expertise in methods that yield significant benefits for human health, sustainability, economic growth, cultural heritage, and overall well-being. Investing in conscientious and sustainable food systems can improve individual and planetary quality of life by preventing diseases, delaying ageing, and enhancing well-being. While healthy eating habits begin at home, schools play a pivotal role in reinforcing them from an early age. Despite progress, challenges remain, underscoring the need for prioritizing food education and literacy across all ages. Knowledge of how sustainable food production impacts personal health and well-being is critical. A holistic approach is essential for addressing these complexities, considering physical, mental, social, and environmental factors to identify balanced and effective solutions. Such analyses examine how system components interact, guiding the development of sustainable practices. The DM4You project exemplifies this approach. It unites Portuguese partners to promote local food consumption, focusing on soups, an integral part of traditional Portuguese cuisine, made with diverse vegetables, legumes, olive oil, and sometimes by-products. DM4You monitors 80 healthy participants over three months to assess dietary habits, focusing on soup and fruit consumption. This crossover study evaluates the influence of diet on health, offering insights into sustainable and health-promoting dietary practices.

## 1. Transforming Food Systems for a Sustainable Future: An Integrated Approach

Food systems (FS) are defined as encompassing the entire spectrum of interconnected value-added activities involved in the production, aggregation, transformation, distribution, consumption, and disposal of food products derived from agriculture, forestry, or fisheries. These systems are situated within the broader economic, social, and natural environments. The food systems comprise several subsystems, including agricultural systems, waste management systems, and production factor supply systems, which interact with other fundamental systems, such as those concerned with energy, trade, and health.

Transformations in the structure of food systems may be precipitated by shifts in the configuration of other systems. To illustrate, a policy promoting organic food production would exert a significant impact on the entire food system. The European Commission has identified organic production as a pivotal component in achieving the objectives set forth in the European Green Deal, the “From Farm to Fork” strategy, and the biodiversity strategy. In consequence of the above, the European Commission has set an ambitious target of converting 25% of EU agricultural land to organic farming by 2030, together with a significant increase in organic aquaculture [1].

A sustainable food system strives to guarantee food security and nutrition for all, while preserving the economic, social, and environmental foundations essential for food security and nutrition, without jeopardizing the capacity of future generations to meet their own needs. The United Nations Sustainable Development Goals (SDGs), adopted in 2015, place sustainable food systems at the core of their agenda. The Sustainable Development Goals (SDGs) set forth a series of transformative changes in agriculture and food systems with the objective of eradicating hunger, achieving food security, and improving nutrition by 2030. To meet these goals, it is necessary to reshape the global food system in a way that makes it more productive, more inclusive of marginalized populations, more environmentally sustainable, more resilient, and capable of providing healthy and nutritious diets for all.

These challenges are complex and systemic in nature, necessitating a combination of interconnected actions at the local, national, regional, and global levels [2]. As emphasized by [3], the unavoidable demographic, economic and climate changes that are expected to occur in the coming decades will significantly exacerbate these issues unless the current systems are reformed. Moreover, food systems were accountable for 34% of the total global greenhouse gas emissions in 2015, thereby emphasizing the imperative for a sustainable transformation [4].

### 1.1. Building Sustainable Food Systems

In this context, innovation is a crucial factor in enabling concerted and coordinated efforts to transition towards healthier, more equitable, resilient, and sustainable food systems. To achieve such a transition, it is necessary to adopt a holistic approach to the analysis of food systems, encompassing the entire spectrum from farm to fork and extending to waste management. It is essential that policy options, incentives, and investments facilitate transformative changes across all domains of food systems, including food production, supply chains, food environments, and consumer behavior. The food systems approach emphasizes the importance of recognizing potential synergies, which necessitates an in-depth understanding of how policies implemented at one stage of the food supply chain can generate ripple effects throughout the entire system. It is of the utmost importance to identify strategic points for intervention along the supply chain to maximize the synergies and optimize the overall benefits [5].

Despite the global food system transitions that have made diets more accessible, there has not been a uniform improvement in nutritional outcomes, environmental sustainability, inclusivity, or equity [6,7]. It is not necessarily the case that increased accessibility equates to improved nutritional quality or the adoption of environmentally and socially sustainable practices.

There remains a clear correlation between access to recommended diets and the level of food system development within a given country, with more industrialized nations enjoying greater accessibility. However, emerging data indicate that in some countries, the consumption of processed foods is prevalent not only among urban households but also increasingly among rural populations [8]. This issue is highly relevant within the framework of the New Urban Agenda, as adopted by the United Nations General Assembly [9].

The impact of the COVID-19 pandemic on food systems, food production, and food security has been extensively documented by various scholars [3,10,11,12,13,14,15], alongside the repercussions of the war in Ukraine [16,17,18]. These events underscore the critical need to “build” more resilient food systems capable of adapting to shocks and disruptions while formulating policies that ensure food security during crises.

The development of targeted support policies can significantly enhance nutrition, reduce food loss and waste, optimize resource utilization, prevent deforestation, curb biodiversity loss, limit greenhouse gas emissions, reduce reliance on harmful chemicals, and provide support to farmers [19,20].

Key measures include incentivizing the adoption of sustainable agricultural practices local consumption, developing local markets with shorter supply chains, and improving rural–urban linkages [21]. Additionally, price regulation strategies that favor producers over intermediaries and efforts to reduce pre- and post-harvest losses and mitigate food waste are crucial components of a comprehensive approach to sustainable food system transformation [12,22].

### 1.2. Strategies for Advancing Sustainable Food Systems and Policy Alignment

The 2030 Agenda has underscored the pivotal role that transforming agri-food systems play in accelerating progress towards achieving numerous Sustainable Development Goals (SDGs). However, it has also highlighted the inherent complexities in driving transformative changes across the global food landscape [22,23,24,25]. According to FAO Director-General QU Dongyu, the uneven recovery from the global pandemic, compounded by the conflict in Ukraine, has disrupted access to nutritious foods and healthy diets [16,17,18]. This evolving “new normal” is characterized by the convergence of climate change, conflicts, and economic instability, pushing populations to the brink of food insecurity. In response, a mere maintenance of the status quo is untenable [26].

Reflecting this urgency, UN Deputy Secretary-General Amina Mohammed, at the UN Food Systems Summit + 2 Stocktaking Moment (UNFSS + 2) held in Rome from 24–26 July 2023, officially closed the summit by presenting the Secretary-General’s Call to Action. This call emphasizes the need for all stakeholders to mobilize and commit to implementing six specific objectives aimed at accelerating the transformation of food systems. These objectives are critical to addressing the multifaceted challenges currently facing global food security:1st—Integrate food system strategies into all national sustainable development policies;2nd—Establish food system governance that brings together all stakeholders in society;3rd—Invest in research, data, innovation, and technological capabilities;4th—Promote the involvement and accountability of businesses in shaping the sustainability of food systems, recognizing their centrality in the food and agricultural ecosystem;5th—Ensure full participation of women, farmers, youth, and indigenous peoples at the local level in all stages of implementation;6th—Ensure long-term and favorable financing, investments, budget support, and debt restructuring [27].

In this context, the need for targeted support policies is imperative, both globally and within more developed nations, including Portugal. The alignment of public policies with European frameworks has resulted in the development of programs, financial support, and assistance for farmers, processors, retailers, and large-scale distributors. However, the responsibility also lies with individuals to make informed decisions that can contribute to challenging and changing the existing status quo [28].

Achieving healthy diets within the framework of sustainable food systems is a complex endeavor that necessitates a multi-dimensional approach. According to the Directorate-General for Food and Veterinary Affairs (DGAV), sustainable nutrition encompasses diets that are culturally accepted, accessible, and meet the physiological needs of the global population [28,29]. These diets must also ensure food safety, respect for biodiversity, and ecosystem integrity, while being economically viable and equitable. Furthermore, they should optimize the use of human and natural resources, actively combat food waste, and support the well-being of present and future generations [30]. To meet these goals, sustainable nutrition (SN) is founded on several core principles. It must

Be accessible to everyone;Be culturally accepted;Meet physiological needs;Be safe;Respect biodiversity and ecosystems;Be fair and economically viable;Utilize human resources effectively;Utilize natural resources efficiently;Combat waste.

Achieving these principles requires a concerted effort to strengthen urban–rural linkages, enhance product design, invest in innovative food systems, implement selective collection for alternative food waste uses, and promote behavior change within food environments. Recently, key performance indicators (KPI) were identified by analyzing legislation, guidelines and scientific literature to measure social sustainability in the Italian agri-food sector [31]. They are offering a set of 24 different KPIs, organized into macro-areas of interest such as employment and training, health and safety at work, human rights, territorial community, and health and safety in production, allowing farms to evaluate and improve their practices in line with social expectations and European policy directives [31].

The establishment and advancement of sustainable food systems must be founded upon a comprehensive and multifaceted approach to Food Education, which is vital for effecting change at multiple levels (Figure 1).

Food education serves as the cornerstone upon which these interactions are built, equipping individuals and communities with the requisite knowledge to make informed decisions about sustainable and nutritious food choices. It is thus imperative that an educational foundation be established to facilitate the creation of a well-informed society that can support and participate in sustainable food systems. This will ultimately result in the creation of healthier and more resilient communities and environments.

A recent study conducted in Istanbul, Turkey, involving 1600 volunteers with an average age of 28.2 ± 10.9 years, corroborates these findings [32]. The objective was to analyze the relationship between sociodemographic and anthropometric variables, nutritional knowledge and nutritional literacy. A total of 94.6% of respondents were deemed to possess adequate nutritional literacy [32]. The results indicated that age, gender, marital status, level of education, professional status, body mass index (BMI), and nutritional education were significantly associated with nutritional literacy [32]. A negative correlation was observed between body mass index (BMI) and age with nutritional literacy, whereas a positive correlation was evident between knowledge about nutrition and nutritional literacy [32].

Health literacy or the ability to access, understand, and use health information, has been recognized as an international public health goal. Nutrition literacy, a specific form of health literacy, has emerged as scholars explore the skills involved in being both food literate and health literate. This perspective suggests that a well-rounded concept of nutrition literacy should integrate core aspects of both health and food literacy [33].

Nutritional behavior is a highly complex phenomenon that is influenced by several factors, including nutritional knowledge. It is well documented that effective and continuous nutrition education plays an important role in the protection and development of health. It is also fundamental in changing the dietary habits of individuals of all ages. In the context of health-enhancing behaviors, health literacy is regarded as a pivotal predictive factor [34]. A lack of health literacy impairs the ability to identify and recognize health issues. Nutrition literacy, which constitutes an essential component of health literacy, denotes the capacity to comprehend, interpret, and access fundamental nutritional information to make informed decisions regarding nutritional matters. It is well established that an individual’s knowledge, attitudes, skills, and behaviors concerning food and nutrition influence their dietary choices. A growing body of evidence indicates that most individuals encounter difficulties in interpreting the information presented on food labels [35]. Those with inadequate health literacy and/or mathematical abilities tend to exhibit poorer health outcomes [35].

## 2. Integrating Culture and Technology for Sustainable Food Systems: A Multidisciplinary Approach

In contemporary society, the food ecosystem is a multifaceted system that integrates environmental, social, economic, and ethical dimensions, reflecting the inter-connectedness of these factors in shaping sustainable and equitable food practices [36]. Achieving sustainable food systems requires a multidisciplinary approach that encompasses a broad spectrum of knowledge, encouraging change and fostering acceptance among all stakeholders while respecting cultural nuances. Cultural factors play a crucial role in determining the acceptance or rejection of various foods, such as insects, certain animals, and plants, which can vary significantly across sociocultural contexts [37]. It is also necessary to consider the lack of data on the chemical composition of (specifically) insect-based foods and toxicity studies, and potential allergenic implications must also be considered when determining whether to accept a foodstuff [38]. Also, we must consider how ethical decision-making within the food chain plays a crucial role in preventing food fraud and ensuring food safety, which in turn has wider benefits for global health and well-being. The reallocation of resources saved on combating fraud to address issues of hunger, oral health, and systemic health can facilitate the development of healthier communities and promote social justice [39]. Ethical conduct within the food chain reflects a responsibility to oneself, to others, and to the greater good, embodying values of compassion and respect for all life. Ultimately, this entails a commitment to human dignity and the interconnectedness of all beings [40].

In primary food production, it is essential to employ techniques and methods that not only respect the environment but also ensure the safety and quality of food products. Sustainable production practices such as organic farming, holistic management, syntropic agriculture, regenerative agriculture, and biodynamic agriculture represent a few of the many approaches that contribute to this goal. Agricultural product certification, such as Global G.A.P., provides a reliable and assured pathway for producers, processors, warehouse operators, and the entire food chain, ensuring that consumers receive safe and high-quality food [41].

In food processing, traditional thermal techniques remain widely used, relying on temperature changes (both high and low) to preserve and transform food while ensuring its safety [42]. However, ongoing research is focused on developing alternative non-thermal methods to achieve the same goals. These include High Pressure Processing (HPP), which allows for cold pasteurization using high pressure, as well as emerging technologies like microwave processing, ultrasonics, and food irradiation, among others [43].

It is imperative that the food industry prioritizes the promotion of sustainably produced foods and the development of strategies that can reduce food waste. This must be achieved by focusing on the protection of the environment and natural resources. To guarantee food safety, security and sustainability, it is essential that food production is increased, that the impact of this production on the environment is evaluated, and that innovative research, data, techniques and perspectives are developed [44].

Food technology encompasses any technological advancements that enhance food production, distribution, and supply, directly impacting how foods are produced, sold, and distributed. Companies are increasingly adopting the latest technologies to create jobs, reduce hunger, and promote responsible production and consumption throughout the entire food cycle [45].

In the domain of animal production, Artificial Intelligence (AI) has shown signific-cant efficacy, particularly in poultry farming [46]. AI systems can detect health issues in birds by analyzing the sounds they emit, and AI-powered robots are being deployed to perform tasks such as egg collection and assisting in the slaughtering process [46]. The objective is to redefine future agriculture. This is being achieved by utilizing Industry 4.0 applications, including artificial intelligence (AI), the Internet of Things (IoT), and cyber–physical systems. These are facilitating monitoring, precision forecasting, and providing comprehensive insights and system optimization, thereby transforming agricultural productivity [47].

The digitalization and integration of advanced technologies like automation, artificial intelligence, and machine learning have also revolutionized the healthcare sector. Patient data analysis enables the creation of personalized treatment plans tailored to individual needs, while large-scale data analysis enhances the quality of medical and epidemiological research [48].

In the field of medicine, immunotherapies are increasingly being used to treat various conditions, including certain cancers and degenerative diseases such as Alzheimer’s. For instance, “Lecanemab”, approved by the US Food and Drug Administration (FDA) in January 2023, is designed to remove amyloid protein buildup in the brains of individuals with early-stage Alzheimer’s disease [49]. This approach signifies a shift from the traditional “one size fits all” model of medicine towards more individualized treatments. Telemedicine has experienced significant growth, providing remote medical care and patient monitoring, which is particularly valuable in remote areas or for patients with limited access to traditional healthcare [49]. However, the widespread adoption of technology in healthcare also presents challenges, including concerns over data privacy, cybersecurity, and the need for adequate training and empowerment of healthcare professionals. It is also critical to ensure that these technologies are applied ethically and in ways that prioritize patient welfare [50].

In biotechnology, there has been an exponential increase in research, yielding valuable insights that are being used to improve medical treatments, agricultural practices, and environmental solutions [51]. Environmental biotechnology, exemplified by bioremediation techniques using microorganisms to degrade plastic or the development of disease-resistant crops, highlights the potential of this field. Nonetheless, issues related to regulation and ethics, such as the use of genetically modified organisms (GMOs) and their release into the environment, necessitate rigorous safety assessments and responsible practices [52,53]. The complexities of modern supply chains and consumer preferences for safe food have driven the development of smart packaging, which encompasses active and intelligent technologies [52,53,54]. This innovation is becoming increasingly significant as it enhances the safety and quality of food products [54].

More recently, in a study on the association between ultra-processed foods (UPFs) and the increased prevalence of obesity and its complications, the authors state that promoting the consumption of whole and minimally processed foods and implementing nutrition education programs in schools are crucial steps to counteract the prevalence of obesity, which has become a global health problem [55]. They also point out that interdisciplinary and intersectoral collaboration will be essential to develop comprehensive solutions and improve public health outcomes worldwide.

The intersection of sustainable development and environmental biotechnology holds significant potential for addressing environmental challenges and fostering a healthier planet for all. This involves meeting present needs without compromising the ability of future generations to meet theirs.

In Portugal, the National Strategy for Food and Nutritional Security (ENSANP) was established, drawing on contributions from various governmental sectors and the results of extensive consultations with relevant entities [56,57]. This initiative also expanded the membership of the National Food and Nutritional Security Council (CON-SANP), initially created by Resolution of the Council of Ministers No. 103/2018, on July 26. This integrated approach underscores the importance of collaborative efforts in ensuring food security and nutritional well-being on a national scale.

### Food Waste

It has been estimated that food losses due to the processing and production operations of the agri-food sector represent between 30 and 80% of overall yield [58]. On 30 October 2023, a pivotal report was presented, highlighting that food waste “occurs primarily at the household level”. This finding underscores the need to reassess the National Strategy for Food and Nutritional Security, which has been traditionally focused on operators but now requires a stronger emphasis on consumer behavior. The shift away from traditional Portuguese eating habits, particularly the Mediterranean Diet (MD), is increasingly evident among the younger population, driven by a lack of awareness regarding the critical role of healthy eating in promoting long-term health and well-being [59].

Currently, over half of the adult population in Portugal is overweight, contributing significantly to the high prevalence of non-communicable diseases such as coronary heart disease, obesity, and diabetes. In the European Union, approximately 87.6 million tonnes of food are lost or wasted annually, with domestic food waste now estimated at twice the previous figures, reaching 74 kg per capita in 2021 [60].

In Portugal alone, around 1 million tons of food are wasted every year, with households responsible for 31% of this total [61]. To address this, it is crucial to implement practices and tools that enable the conversion of wasted food into consumable products. The European Union has committed to halving per capita food waste by 2030, supported by policies and instruments introduced in 2020 under the Circular Economy Action Plan, the Farm to Fork Strategy, and the Biodiversity Strategy [62].

On September 29th, on the International Day of Awareness of Food Loss and Waste, at a conference organized by the FAO, Andersen stated the following: “Progress towards each of the Sustainable Development Goals (SDGs) is slowing or regressing. But if there’s one thing we know, it’s that crises can bring out the best in people. We must all show these qualities now—particularly those who have the power to effect change in areas of great impact, such as agri-food systems” [63].

Regarding the importance of agri-food systems in the setbacks we are witnessing, the same author states the following: “We need a sustainable transition of the agri-food system that provides healthy and nutritious food for all [61,62]. That slows climate change and strengthens resilience. That improves livelihoods. And that protects the health of people, animals, and ecosystems. Halving food waste and losses, as advocated in SDG target 12.3, would bring enormous gains: Food security, Climate change mitigation, Protection of nature and biodiversity, a lighter burden on water—which is crucial in a world facing growing climate disruption” [64].

The global food system is currently confronting a series of formidable challenges, including biodiversity loss, depletion of natural resources, food insecurity, climate change, malnutrition, and other health-related issues [65].

A literature review encompassing 160 studies on market dynamics, information dissemination, regulatory frameworks, and incentives aimed at improving food systems, revealed key insights: (i) Less intrusive policy instruments, such as information-based strategies and behavioral nudges, are more popular but less effective compared to more stringent market-based and regulatory measures; (ii) consumers tend to rely on information-based tools to make sustainable food choices and are willing to pay a premium for sustainable products; (iii) sociodemographic factors, particularly gender (with females showing greater awareness) and education level (with higher education correlating with more sustainable choices), play a critical role in shaping these decisions [66].

There is a growing social interest in the interplay between food and health, which increasingly influences food choices. However, this interest is not uniform across society-ty, varying significantly with the economic capacity of individuals or households. This variation is more pronounced among higher economic strata, where awareness and education are generally more advanced, as corroborated by the study [66].

The successful achievement of these goals hinges on the dissemination of knowledge and the promotion of food literacy, an essential concept for informed decision-making. Food choices, which shape consumption patterns, are highly individualized and, in the case of children, are often influenced by family feeding models [67].

For adolescents, the systematic implementation of food and nutrition education in schools is crucial. Such education supports adolescents in making decisions that not only safeguard their current health but also ensure long-term well-being [68]. It has been observed that improving the quality of students’ diets can enhance both their lifestyle behaviors and academic performance, with reciprocal benefits [68].

A study further explores the association between diet quality and lifestyle behaviors among higher education students, emphasizing the need to establish healthy campus committees to plan targeted interventions and educational programs focused on diet and lifestyle [69].

Moreover, there is a positive relationship between sustainable consumption and purchase intentions among higher education students. This indicates that environmental concern can indeed translate into effective purchasing decisions when aligned with product characteristics [70].

## 3. Promoting Healthy Eating in Portugal: National Strategies and the Role of the Mediterranean Diet

In Portugal, the National Plan for Healthy Eating (PNPAS) was established as a central platform for disseminating knowledge on nutrition and healthy eating. Functioning under the Directorate-General of Health (DGS), PNPAS serves as both an information hub and an activation platform, engaging a broad range of stakeholders including the health sector, local authorities, international institutions, associations, social projects, and consumers [71].

Complementing this initiative, the Integrated Strategy for the Promotion of Healthy Eating (EIPAS) is designed to ensure the implementation of its defined measures across various state services and bodies, both direct and indirect [72]. The monitoring and evaluation of these measures are overseen by an interministerial Working Group led by the DGS, ensuring a coordinated approach to advancing public health nutrition.

Recently, PNPAS, in collaboration with the European region and the World Health Organization (WHO), assessed Portuguese children’s exposure to digital marketing of food and beverages [73]. The study revealed that girls aged 13–16 from lower socioeconomic backgrounds are the most exposed demographic, with 81% of foods advertised to children through digital platforms being unhealthy [73].

Portugal benefits from numerous national and European support structures aimed at enhancing food literacy. Among these, the Mediterranean Diet stands out as a historically rooted dietary pattern, recognized as the best diet globally by U.S. News and World Report for six consecutive years [74]. It has also been classified as an Intangible Cultural Heritage of Humanity by UNESCO since 2013 [74]. In support of this, the Mediterranean Diet Competence Center was established in Tavira in 2018, followed by the formation of the Network of Higher Education Institutions for the Safeguarding of the Mediterranean Diet (RIESDM) in 2019, to which we belong, alongside the Regional Council for the Mediterranean Diet [75].

The Mediterranean Diet, structured around 10 core principles, serves as a guide for consumers to adopt healthier and more sustainable eating habits, considering the three pillars of sustainability—economic, environmental, and social [76,77]. A recent study by the University of South Australia highlights the diet’s economic benefits, demonstrating that a family of four can save $28 per week, or $1456 annually, by adhering to the Mediterranean diet compared to a typical Western diet [78]. Moreover, studies conducted in the UK have shown that following the Mediterranean diet is associated with a reduced risk of dementia, even among individuals with a genetic predisposition [78].

These findings underscore the significance of integrating traditional dietary practices like the Mediterranean Diet into national strategies, not only for improving public health but also for promoting sustainability and economic resilience [78].

### DM4You

DM4You—Potential of the Mediterranean Diet in Increasing Quality of Life: +sustainability + health is led by the Polytechnic Institute of Portalegre and involves 12 other national partners as well as some international partners.

With a duration of 3 years, the project aims to be a driving force to reinforce, strengthen, disseminate, and educate about the importance of the principles of the Mediterranean Diet, specifically the consumption of soup and fruit and their effects on the health and well-being of populations. In addition to training and food education, the project embraces a more ambitious goal: it proposes to study the effects of soup consumption, through a clinical trial, on inflammatory processes associated with aging. In addition to blood, saliva collection may also allow for the identification of protein biomarkers in a less invasive manner, facilitating the continuation of the studies.

## 4. Conclusions

The promotion of sustainable food systems and healthier dietary habits requires a multifaceted approach encompassing education, technological innovation, cultural integration, and policy alignment. By focusing on holistic strategies, such as those demonstrated by the DM4You project, it is possible to enhance public awareness and encourage the adoption of practices that are both health-promoting and environmentally sustainable.

The DM4You initiative illustrates the potential of traditional dietary elements, such as soups and fruits, in improving well-being and reducing the impacts of ageing, while respecting cultural heritage. These efforts align with broader objectives like reducing food waste, combating climate change, and fostering resilient food systems capable of withstanding crises.

Moreover, integrating food education into public health initiatives and fostering literacy in nutrition and sustainability across all societal levels are critical for empowering informed food choices. The adoption of evidence-based practices and policies, supported by collaborative efforts, is essential to overcome current challenges.

Ultimately, transforming food systems requires sustained commitment at local, national, and global levels, ensuring that they are equitable, inclusive, and resilient for present and future generations. By prioritizing sustainability and well-being, it is possible to create food systems that serve as a foundation for healthier lives and a more sustainable planet.

## Figures and Tables

**Figure 1 foods-13-03829-f001:**
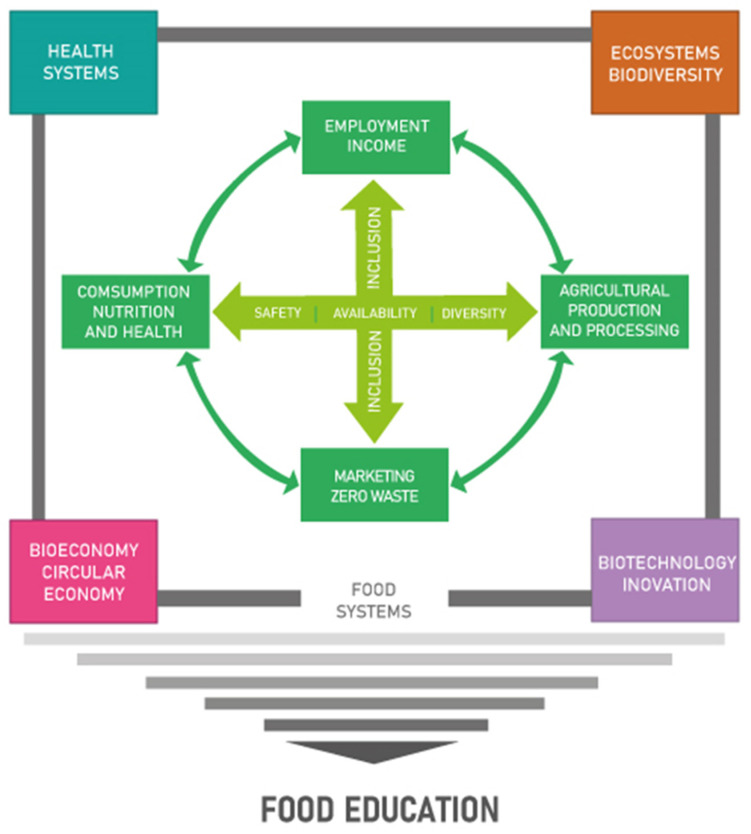
Food Education as Support and Foundation for Development (Adapted from [23]).

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
