# Peer review of "Advancing Health and Sustainability: A Holistic Approach to Food Production and Dietary Habits"

_foods, 2024, doi:10.3390/foods13233829_

Round 1
Reviewer 1 Report
Comments and Suggestions for Authors
The topic deserves the attention.
Content of the paper should be more in line with the proposed title.
Authors list must be corrected. Is there any author after and, and before 5?
Graça P. Carvalho 1,2, Eduardo Costa-Camilo 1, 3, Isabel Duarte 4 and 5
Abstract should be shortened, and rewritten accordingly.
Page numbers and Text lines are not included, so it is hard to point out comment to the specific line/page number.
Page 5: "Cultural factors play a crucial role in determining the acceptance or rejection of various foods, such as insects, certain animals, and plants, which can vary significantly across sociocultural contexts (Luderer & Calado, 2023)..."
By my opinion not just cultural factors are crucial. There is a lack of data about chemical composition of (specifically) insect-based foods, the toxicity studies, and potential allergenic implications.
All above must be considered in order to have safe food.
Author Response
Please see the attacment.

Reviewer 2 Report
Comments and Suggestions for Authors
The manuscript aims to contribute to the literature on sustainable food systems and the role of diet in health and environmental sustainability. However, several fundamental issues compromise the manuscript’s quality and scientific rigor. These issues span from structural and conceptual weaknesses to deficiencies in methodology, analysis, and relevance of literature.
Major Issues
- The manuscript largely reiterates known information and general ideas prevalent in the existing literature without providing new empirical data, innovative methodologies, or a unique theoretical perspective. This lack of novelty fails to meet the standard of contribution required for a research article in a scientific journal.
- The manuscript lacks sufficient detail on the research methods and data analysis used in the study. For a manuscript that claims to adopt a holistic approach to food systems, it is crucial to outline specific methodologies, participant selection criteria, and data analysis techniques. The absence of these details makes it impossible to assess the validity and reliability of the research findings.
- The manuscript does not clearly define key concepts and terms, such as “holistic approach,” “sustainable food systems,” and “healthy eating habits.” It also fails to establish a clear theoretical framework guiding the study. This lack of conceptual clarity undermines the academic rigor of the paper.
- Much of the text is repetitive, with the same ideas presented multiple times in different sections without additional depth or insight. This redundancy not only detracts from the overall coherence and flow of the manuscript but also unnecessarily extends the length without adding substantive content.
- The manuscript does not adequately integrate or critically engage with current literature. References are often outdated or tangentially related to the manuscript’s core topics. Additionally, the literature review lacks a critical analysis of previous studies, failing to build a compelling case for the study’s relevance and importance.
- The manuscript is poorly organized, with sections and subsections not logically flowing from one to another. The lack of a structured argument or clear narrative thread makes it difficult for readers to follow the manuscript’s main points and contributions.
Minor Issues
- The manuscript contains numerous grammatical and stylistic errors that impede readability. These issues suggest that the manuscript did not undergo thorough proofreading or professional editing.
- Figures and tables included in the manuscript are poorly presented and inadequately described. There is a lack of clear legends or explanations, making it difficult for readers to understand their relevance or interpret their content accurately.
Given the above issues, the manuscript does not meet the standards of a scientific journal focused on sustainable food systems and health. The fundamental flaws in originality, methodology, theoretical grounding, and overall presentation significantly hinder its contribution to the field.
Comments on the Quality of English Language
The manuscript contains numerous grammatical and stylistic errors that impede readability. These issues suggest that the manuscript did not undergo thorough proofreading or professional editing.
Round 2
Reviewer 2 Report
Comments and Suggestions for Authors
The revised manuscript can be accepted for publication.
